# Management of Scar Contractures of the Hand—Our Therapeutic Strategy and Challenges

**DOI:** 10.3390/jcm13051516

**Published:** 2024-03-06

**Authors:** Hoyu Cho, Shimpei Ono, Kevin C. Chung

**Affiliations:** 1Department of Plastic, Reconstructive and Aesthetic Surgery, Nippon Medical School Hospital, 1-1-5 Sendagi Bunkyo-ku, Tokyo 113-8603, Japan; hoyu-cho@nms.ac.jp; 2Section of Plastic Surgery, Department of Surgery, The University of Michigan Health System, Ann Arbor, MI 48105, USA; kecchung@med.umich.edu

**Keywords:** hand, scar contracture, dimensional classification, skin defect, limb reconstruction

## Abstract

The essence of treating scar contractures lies in covering the skin deficit after releasing the contractures, typically using flaps or skin grafts. However, the specific characteristics of scar contractures, such as their location, shape, and size, vary among patients, which makes surgical planning challenging. To achieve excellent outcomes in the treatment of scar contractures, we have developed a dimensional classification system for these contractures. This system categorizes them into four types: type 1 (superficial linear), type 2-d (deep linear), type 2-s (planar scar contractures confined to the superficial layer), and type 3 (planar scar contractures that reach the deep layer, i.e., three-dimensional scar contractures). Additionally, three factors should be considered when determining surgical approaches: the size of the defect, the availability of healthy skin around the defect, and the blood circulation in the defect bed. Type 1 and type 2-d are linear scars; thus, the scar is excised and sutured in a straight line, and the contracture is released using z-plasty or its modified methods. For type 2-s, after releasing the scar contracture band, local flaps are indicated for small defects, pedicled perforator flaps for medium defects, and free flaps and distant flaps for large defects. Type 2-s has good blood circulation in the defect bed, so full-thickness skin grafting is also a suitable option regardless of the defect’s size. In type 3, releasing the deep scar contracture will expose important structures with poor blood circulation, such as tendons, joints, and bones. Thus, a surgical plan using flaps, rather than skin grafts, is recommended. A severity classification and treatment strategy for scar contractures have not yet been established. By objectively classifying and quantifying scar contractures, we believe that better treatment outcomes can be achieved.

## 1. Background

Scar contractures, which are characterized by reduced joint motion due to scarring, are distinct from joint contractures where the joint itself is contracted. These contractures commonly occur on extremities and are notably prevalent over joints with long-axial scars. The fundamental pathology of scar contractures is a skin deficit, which becomes evident upon releasing the contracture (Figure 1). Treatment involves releasing these contractures and subsequently covering the exposed defects with flaps or skin grafts. Given the variability in the location, shape, and size of scar contractures among individuals, selecting an appropriate surgical procedure can be challenging. To enhance both functional and aesthetic outcomes in treating these contractures, we propose a dimensional classification system as a foundation for treatment decisions. This paper aims to present our treatment strategy, underpinned by this dimensional classification of scar contractures.

## 2. Assessment of Scar Contractures

The accurate identification of the etiology of a contracture is essential for its effective treatment planning. The causes of contractures can be classified into two categories: extra-articular (including scar contracture, tendon adhesion, and muscular contracture) and intra-articular (including restrictions of joint supporting tissues, shrinkage, and articular surface destruction) (Figure 2). Notably, certain cases may present with both extra- and intra-articular elements. For example, severe trauma may cause deep lesions extending from the skin to the joint, or joint contractures may develop secondary to severe scar contractures. It is imperative to avoid situations where the assumption that a contracture is solely skin-related, which leads to an initial plan of scar contracture release followed by skin grafting, only to find intraoperatively that the contracture extends deeper than anticipated, necessitating a shift to flap surgery. Therefore, thorough preoperative diagnosis and evaluation are essential. To distinguish between scar contracture and joint contracture, two diagnostic checkpoints are proposed. First, in the case of a proximal interphalangeal (PIP) joint contracture, if the condition is a scar contracture, passive extension of the PIP joint will result in the scar turning pale (Figure 3). During surgery, the area where the skin contracture is strongest should be released. Second, the adjacent metacarpophalangeal (MCP) joint is placed in a flexed position, and the skin on the palm side is loosened, then the PIP joint is passively extended. In scar contractures, this maneuver facilitates the easy extension of the PIP joint. Conversely, in cases of joint contracture at the PIP joint, passive extension remains unachievable even with the skin loosened (Figure 4).

## 3. Surgical Indication and Timing

Scar contractures of the hand present various symptoms that can limit hand function, and the definition of the functional disorder may differ for each patient. This definition varies from one individual to another, based on the patient’s personal and occupational characteristics, and includes an aesthetic perspective. Symptoms range from a mild feeling of tightness with slight functional loss to severe limitations in the range of motion, including degradation of tendon gliding and joint contractures. There is no doubt that severe cases requiring improvement in both function and aesthetics are candidates for surgery. On the other hand, recent patient satisfaction studies have reported that improvements in hand aesthetics increase patient satisfaction [1,2,3]. In cases of hand scar contractures, surgery to enhance aesthetic appearance may be useful even in mild cases with little functional impairment. It is hoped that future research will establish evidence supporting this approach.

After traumatic skin defects and fresh burns have healed and epithelialized, it is essential to start joint range of motion exercises at an early stage to prevent joint contractures, ideally under the guidance of hand therapists. Splints, particularly at night, are used to prevent scar contractures. The application of compression garments and steroid tape to maturing scars to minimize hypertrophy can be beneficial [4]. It is preferable to wait for scars to mature before surgery. Normally, scar maturation requires time, and it may take six to twelve months before surgery is advisable [5,6,7]. However, in cases of severe scar contracture with functional limitation where there is a tendency to develop joint contractures, surgery should be performed immediately. From an age perspective, children are more likely to develop contractures. Digital flexion contracture, especially the long-term fixation of the PIP joint in a flexed position, impairs the digital extension mechanism, so early surgical intervention is required once the wound has healed and contracture develops. In children, contracture recurrence may occur after scar contracture formation as they grow older. The timing of re-operation must be determined carefully, taking into account the degree of contracture and the child’s growth stage.

## 4. Treatment Strategy for Scar Contractures of the Hand

The essence of scar contracture is a lack of skin. Therefore, when the scar contracture is released, the resulting skin deficit appears as a defect. It is necessary to reconstruct this defect both functionally and aesthetically to prevent the recurrence of scar contracture. In the treatment of hand scar contractures, reconstruction with flaps using local healthy skin, local flaps, or pedicled perforator flaps is generally preferable to skin grafts. With skin grafts, secondary contraction is always an issue from a long-term perspective, and future shrinkage of the transplanted skin graft may lead to the recurrence of scar contracture. Particularly, the thinner the skin graft the more likely it is to undergo secondary contraction [8,9], so full-thickness skin grafting is usually recommended for treating scar contracture. Furthermore, when skin is harvested from areas other than the hands, like the groin or lower abdomen, and grafted, the color and texture match are often poor, resulting in a patchwork-like appearance. Conversely, when a flap is inserted into a defect after releasing the scar contracture the flap is gradually stretched in the longitudinal direction by stretching stimulation [10], reducing the likelihood of scar contracture recurrence. By choosing local flaps and pedicled perforator flaps, reconstruction using similar tissue around the defect becomes possible, resulting in highly satisfactory outcomes in both function and aesthetics. Additionally, opting for a flap over a skin graft eliminates the need for rest during tie-over fixation, allowing rehabilitation to commence early post-surgery. Furthermore, even if tenolysis or joint mobilization is required in the future, the final treatment outcome is often better if the overlying skin has been reconstructed with a flap.

However, flap surgery can only be used when the defect is small to medium in size and healthy skin remains around the defect (Figure 5). When selecting a reconstruction method for a defect after scar contracture release, the following three criteria should be considered: (A) the size of the defect, (B) the presence or absence of healthy skin around the defect, and (C) the blood flow in the defect bed. Firstly, the defect size is assessed as small, medium, or large. In this context, the absolute measured value (cm) of the defect is less important than its anatomical location. For instance, a two cm defect on the hand is considered small, whereas the same two cm defect on a digit is considered large.

### 4.1. Small-Size Defect

If healthy skin remains around the defect, simple sutures, z-plasties, or conventional local flaps are the first choices for small-size defects. After suturing in a straight line, one or several small z-plasties can be added. If it is difficult to release the contracture sufficiently with z-plasties alone, a local flap should be considered. Particularly effective is a technique where a transposition flap is rotated 90 degrees and inserted into the defect after dividing the scar contracture line. In both cases, the contracture line is divided by sandwiching normal skin between the skin on either side of it.

### 4.2. Medium-Size Defect

For medium size defects with healthy surrounding skin, a pedicled perforator flap (or regional flap) is selected. Pedicled perforator flaps are particularly useful as a minimally invasive surgical technique that does not sacrifice the main artery. This technique involves covering the defect by rotating a skin island, like a propeller, around a perforator near the defect [11]. The perforator flap can stabilize blood supply by containing the perforator and can expand the coverage range through rotational movement. The perforator pedicled propeller flap is especially useful for limb defects where donor sites are limited. This flap can cover a defect with a 180-degree rotation and can be harvested from the longitudinal axis of the limbs [11].

### 4.3. Large-Size Defect

For large defects, where it is difficult to cover the defect with local skin transfer, free flaps or distant flaps are selected.

On the other hand, if there is no healthy skin around the defect, such as in the case of scar contracture after whole-body burns, the first choice is skin grafting, regardless of the defect size (Figure 5). It is important to note that skin graft survival depends on the blood circulation from the underlying bed, so it is not suitable for wounds where important structures with poor blood flow (such as tendons, joints, bones, etc.) are exposed. In many cases of scar contracture, there is usually no problem with the underlying blood flow. However, if the scar is deep (as in cases of severe trauma extending to soft tissues) or if the surrounding blood circulation is poor (such as after radiation therapy), free flaps or distant flaps should be selected.

## 5. Treatment Choices Based on Dimensional Classification of Scar Contractures

### 5.1. Dimensional Classification of Scar Contractures

To facilitate treatment planning, scar contractures can be morphologically divided into one to three dimensions (Figure 6). Superficial scar contractures are categorized into two types: superficial linear (type 1) and planar scar contractures confined to the superficial layer (type 2-s). Deep scar contractures can be classified into two types: deep linear (type 2-d) and planar scar contractures that reach the deep layer, i.e., three-dimensional scar contractures (type 3). Type 1 is commonly caused by minor cut injuries, type 2-s by second-degree burns, type 2-d by surgeries near joints, and type 3 by severe trauma such as press injuries. Type 3 may be complicated by joint contractures in addition to scar contractures. To distinguish the dimensional classification, preoperative evaluation by actual palpation is important. When the scar reaches deep enough to cause adhesions to the underlayers, the range of motion will be limited. Therefore, it could be possible to estimate the range of the scar by using passive extension, flexion, and pinching, as shown in Figure 3. Scar tissue does not reach flat; instead, there is a mixture of areas that reach superficial layers and areas that extend deeper. However, this is considered to be a simple procedure for determining the surgical strategy.

### 5.2. Scar Contracture Releasing

If there is sufficient healthy skin around the scar contracture, remove as much of the scar as possible. Conversely, if there is not enough healthy skin around the scar contracture, it is advisable not to forcibly remove the scar. The scar contracture line, running along the longitudinal direction, is divided at the part where the contracture is strongest (Figure 1). Particularly when releasing a digital palmar contracture, to prevent a recurrence, the dorsal edge of the skin graft or flap should coincide with or extend beyond the midlateral line in the dorsal direction (Figure 7). Moreover, suture lines in the longitudinal direction that are perpendicular to the skin creases on the palmar side of the digits and hand can cause hypertrophic scars and recurrence of scar contracture. Therefore, scar contracture can be prevented by positioning the skin so that the suture line forms a zigzag in this area or by adding a small z-plasty where the suture line crosses the skin crease (Figure 8). Another critical point is to ensure that no fibrous tissue that is causing the scar contractures remains, not only in the superficial layer but also in the deep layers, and so it should be removed until healthy fat tissue is exposed. During this procedure, fibrous tissues should be carefully dissected or excised under a magnifying loupe to preserve the neurovascular bundle and avoid unnecessary release of the tendon sheath. It is important to remember during surgery that if the scar tissue causing the contracture is not completely removed, the invasiveness of the surgery might exacerbate the scar contracture.

### 5.3. Treatment Policy for Each Type of Scar Contracture

In type 1 (superficial linear scar contracture) (Figure 9) and type 2-d (deep linear scar contracture) (Figure 10), the scar is excised, and simple sutures are performed in a straight line with a z-plasty added on the skin crease due to the availability of plenty of healthy skin around the scar. The purpose of the z-plasty is to incorporate healthy skin into the contracture line from both sides [12]. It is recommended that the z-plasty be performed in a healthy area, not a scarred one. The angle of the z-plasty is usually designed at 60 degrees. The length that can be extended with one z-plasty is approximately 1.7 times the length of one side of the Z. Therefore, if the scar contracture is severe, multiple z-plasties are added. Increasing the length of one side of the Z does not proportionally increase the lengthening effect and may make the surgical scar and skin distortion more noticeable. The appropriate length of one side of a z-plasty is approximately 5 to 8 mm for the digits and 10 to 15 mm for the hand. In cases with severe scar contracture, surgical methods with a higher lengthening effect, such as multiple z-plasty, 4 flap z-plasty, 5 flap z-plasty, or the square flap method, are recommended.

In type 2-s (planar scar contracture confined to the superficial layer) (Figure 11), treatment choices should be selected based on the size of the scar contracture. In small-size defects, primary suturing and z-plasty or a conventional local flap are recommended. For slightly wide band-shaped scar contractures, it is effective to divide the contracture line and insert a transposition flap rotated 90 degrees into the resulting defect. For medium-size defects, a pedicled perforator flap is preferred. Given the tapered structure of the upper limbs, a technique that rotates the skin island in a propeller-like manner from the proximal side, where there is excess skin, to the distal side, where the defect is located, can be applied. Another option is to cover the joint area with a local flap or pedicled perforator flap and cover areas other than the joint with a skin graft. In type 2-s, as the scar is localized in the superficial layer, blood circulation in the defect bed is generally good after scar contracture release. Therefore, full-thickness skin grafting can also be a useful option.

Type 3 (planar scar contractures that reach the deep layer, i.e., three-dimensional scar contractures) is the most severe in the classification of scar contractures (Figure 12). Upon release of scar contracture, important structures with poor blood circulation (tendons, joints, bones, etc.) are likely to be exposed, making skin grafting a less favorable option. A primary linear suture or a local flap is preferred for small-size defects, a pedicled perforator flap for medium-size defects, and a free flap or distant flap for large-size defects.

The above explanation focused on flap reconstruction, but, as shown in Figure 5, skin grafting is indicated regardless of the size of the defect, as long as there is good blood flow in the defect bed. Furthermore, as described earlier, a surgical method in which only the exposed joints and important structures are covered with a flap, and the rest is covered with skin grafts, is also useful. For scar contracture, skin grafting typically involves full-thickness grafts to minimize secondary contraction. When grafting for defects on the palmar side of the digits or hand, it is ideal to harvest similar tissue from the palmar side of the hand or plantar area, but there are limits to the amount of skin that can be harvested. An example of a donor site for skin grafting is the thick split-thickness skin from the thenar, hypothenar, or plantar areas, which can be harvested using a skin knife [13]. Another method involves harvesting full-thickness skin in a spindle shape so that the suture line aligns with the palmar skin crease, the midlateral line of the hand [14], and the submalleolar crease of the foot. If the scar contracture is widespread, skin grafts from the groin region should be selected because, although the tissue will no longer be similar, a large skin graft can be harvested and the surgical scar will be hidden by clothing.

After determining the size of the defect, the skin graft is harvested. The pinch test is used to confirm the width of the skin graft that can be sutured. Ideally, the defect should be covered with a single skin graft, as multiple skin grafts will result in scarring at each edge. Equable tie-over pressure is key to successful skin grafting. Employing approximately 1 cm interval suturing thread and anchoring sutures helps to avoid graft floating. Anchoring sutures are also effective in preventing hematoma and seroma under the skin graft.

In cases of circumferential or multiple and extensive burns of the extremities and joints, the area of scar contracture is large and there are few surrounding healthy donor-site to cover the defect after releasing the scar. The paucity of donors is particularly problematic in pediatric cases. The risk of recurrence of a scar contracture is inversely proportional to the dermis amount, and the wound bed condition is closely related to the quality of the treatment of the scar contracture [15,16]. The use of an artificial dermis for wound bed preparation before skin graft application provides good outcomes in terms of reducing the recurrence rate of scar contractures and increasing the range of motion of the joints [16,17,18].

In any case, it is preferable to approach the surgery with the assumption that the situation is one rank more severe than diagnosed. This is because, if the preoperative evaluation led the surgeon to believe it was a type 1 contracture and they released the scar which turned out to be type 2 with a wider defect this could cause further complications.

### 5.4. Postoperative Management

After releasing a scar contracture, the target joint should be immobilized. External fixation, such as a cast or aluminum splint, is often used, but temporary immobilization with Kirschner wire is especially applied in children’s cases to maintain the correct position. The duration of Kirschner wire insertion should be kept to a minimum to avoid joint contracture, and the wire should be removed within one to two weeks post-surgery. Immediately after the surgery, the limb should be cooled and elevated, and this elevation should be maintained for one to two weeks postoperatively. The treated joints should be immobilized to enhance wound healing and gentle active mobilization of movable joints should be started early after surgery to promote lymphatic and venous return and prevent hand edema. After confirming surgical wound healing, usually one to two weeks postoperatively, rehabilitation of the target joint should be initiated, ideally with hand therapist intervention. Specifically, active and passive mobilization of the joints should be performed frequently (about 5 min per hour) without causing pain to the patient. The patient’s limb should be immobilized with a splint except during rehabilitation until two to three weeks after surgery. After this period, immobilization should only be applied at night, allowing unrestricted use during the daytime. Nighttime immobilization should continue for at least three months postoperatively.

When even the slightest induration of the scar is noted, the nighttime application of steroid tapes should be started immediately. Nighttime application is recommended as the tape may easily come off during hand movements, although daytime application is also possible. In cases of severe edema of the digits, a self-adherent wrap bandage is used. It is important to emphasize that postoperative therapy is as crucial as the surgery itself in the treatment of scar contractures.

## 6. Conclusions

The treatment strategies for scar contractures of the hand are outlined. The hand, having many joints, is prone to scar contractures, particularly with long-axial scars over joints. When reconstructing a scar contracture, it is necessary to consider (A) the defect size, (B) the presence of donor skin around the defect, and (C) the presence of vascular microcirculation in the defect wound bed. Then, an appropriate treatment can be selected based on the dimensional classification of scar contractures.

## Figures and Tables

**Figure 1 jcm-13-01516-f001:**
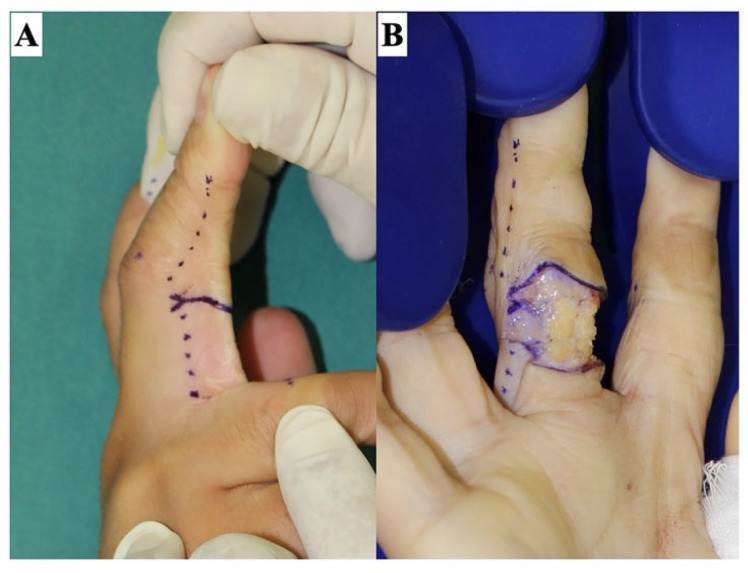
Release of a scar contracture line on a finger. Releasing the scar contracture line causes a skin defect. The stronger the scar contracture, the larger the defect size. The skin incision is made slightly beyond the midlateral line (dotted line), and the Y-shaped incision creates a zig-zag wound edge to prevent re-contracture. The scar tissue in the defect bed is sufficiently dissected and excised until healthy fat tissue is identified. (**A**) The design of the skin incision line. (**B**) After releasing the scar contracture line.

**Figure 2 jcm-13-01516-f002:**
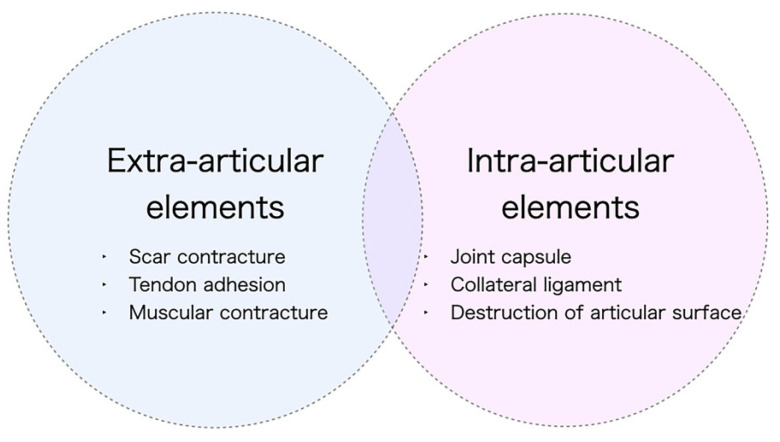
Cause of contracture. Contractures can be broadly classified into cases where the cause is outside the joint (extra-articular element) and cases where the cause is within the joint itself (intra-articular element). Scar contracture is caused by an extra-articular element, the skin.

**Figure 3 jcm-13-01516-f003:**
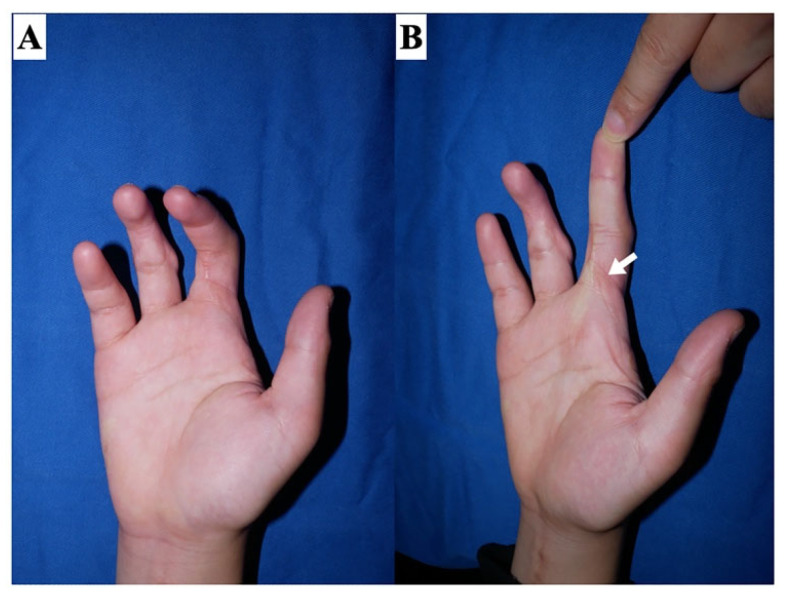
Clinical finding of scar contracture. In scar contracture, when the joint is passively extended (passively stretching the skin), the scar becomes pale (white arrow). (**A**) Relaxed position. (**B**) Finger with a scar contracture in passive extension.

**Figure 4 jcm-13-01516-f004:**
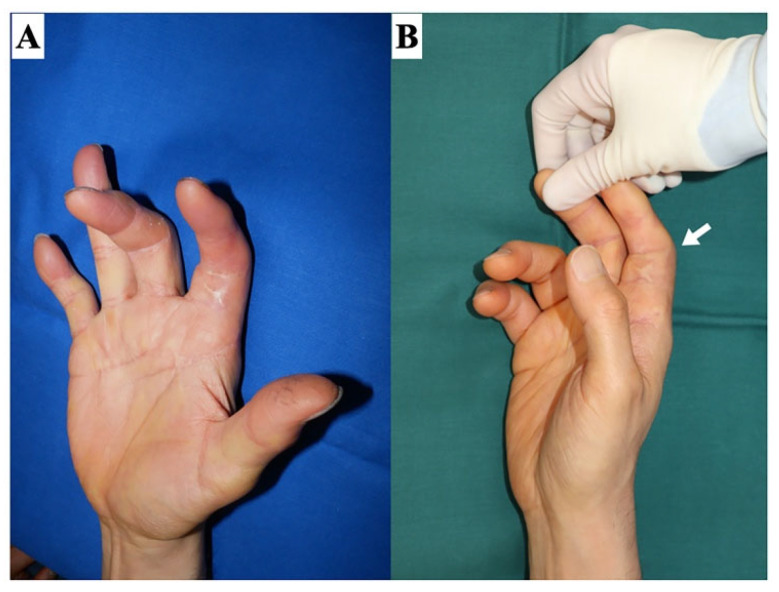
Clinical finding of joint contracture. In joint contractures of the PIP joint, even if the skin is loosened, the PIP joint cannot be extended either actively or passively. (**A**) Relaxed position. (**B**) The PIP joint cannot be extended (white arrow).

**Figure 5 jcm-13-01516-f005:**
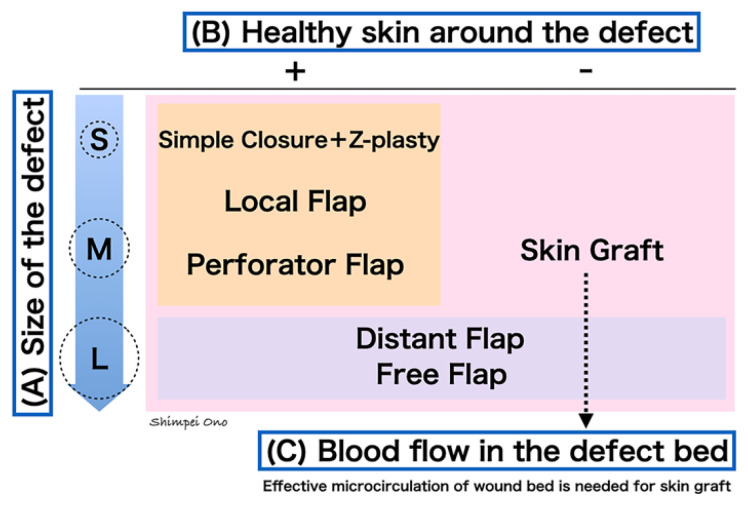
A treatment strategy for hand scar contracture. To decide on the treatment strategy for defects after releasing the scar contracture, there are three judging points: (**A**) the size of the defect, (**B**) the presence of healthy donor skin around the defect, and (**C**) the condition of the vascular bed. S, Small-size; M, Medium-size; L, Large-size.

**Figure 6 jcm-13-01516-f006:**
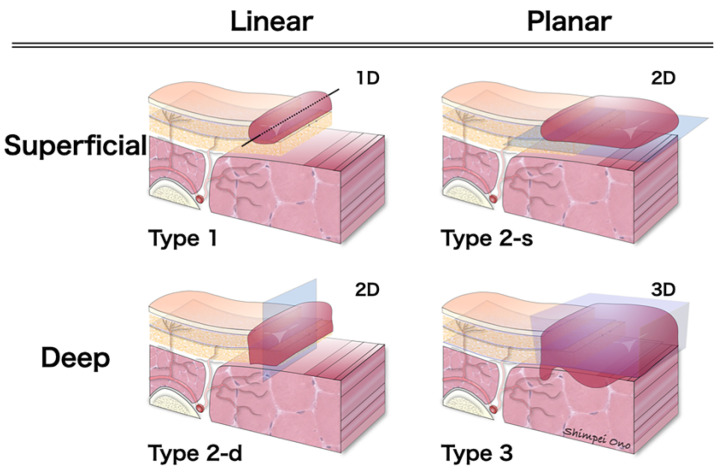
Dimensional classification of scar contracture. Scar contractures are morphologically divided into superficial and deep, linear and planar, and are classified as follows: Type 1 (superficial linear scar contracture), Type 2-d (deep linear scar contracture), Type 2-s (planar scar contracture confined to the superficial layer), and Type 3 (planar or three-dimensional scar contracture reaching deep layers).

**Figure 7 jcm-13-01516-f007:**
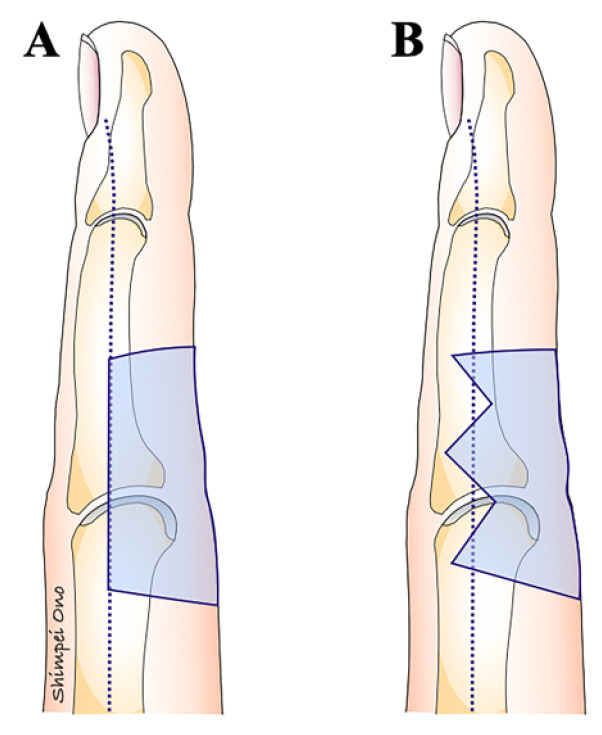
The shape of the defect after the release of a scar contracture on the palm side of the digit. If the defect edge matches the midlateral line or crosses it to the dorsal side, it will be difficult for the grafted skin to undergo secondary contraction. (**A**) The dorsal edge of the donor aligns with the midlateral line of the digit. (**B**) The dorsal edge of the defect extends the midlateral line of the digit with a zig-zag shape.

**Figure 8 jcm-13-01516-f008:**
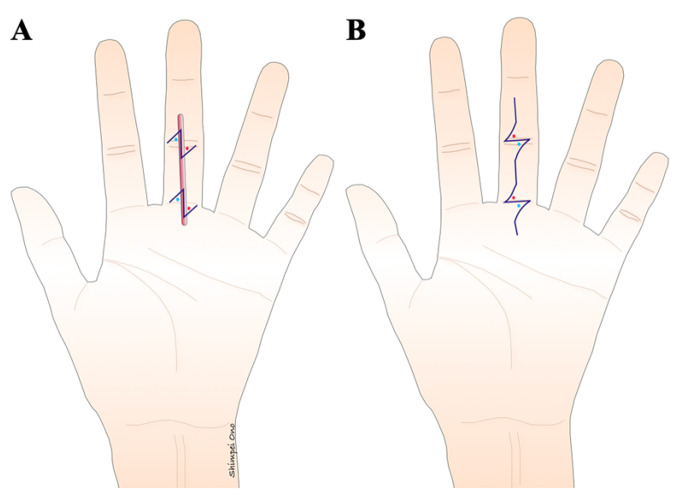
The design of a z-plasty for a linear scar that is perpendicular across the creases of the digit. (**A**) A scar that is perpendicular to the digital crease tends to cause scar contracture. Therefore, the z-plasty needs to be oriented on the crease. (**B**) The z-plasty is designed so that the lateral limbs match the digital crease after the flap is transposed.

**Figure 9 jcm-13-01516-f009:**
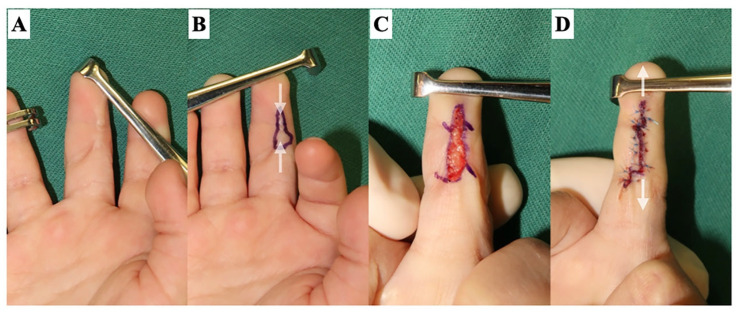
Type 1: superficial linear scar contracture. A 3-year-old male with a history of contact burn. (**A**) The scar contracture of the long finger. (**B**) A linear scar contracture in the longitudinal direction extending from the DIP to the PIP joint on the palm side of the long finger can be identified (white arrow). (**C**) The design of z-plasties over the skin crease after releasing the scar contracture. (**D**) The postoperative scar extends vertically.

**Figure 10 jcm-13-01516-f010:**
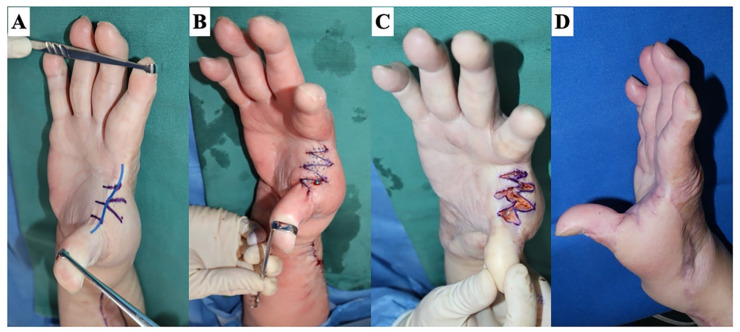
Type 2-d: deep linear scar contracture. A 49-year-old woman underwent reconstruction with a radial artery perforator flap for burn scar contracture on the dorsoradial side of the hand and suffered from scar contracture due to the flap margin that coincided with the first web space. (**A**) The design of a 5 flap z-plasty. (**B**) If the flap is designed appropriately, the flap will naturally move to the desired position once the skin incision is made and the contracture is released. (**C**) Immediately after suturing. (**D**) Six months after the surgery: the abduction of the thumb was enlarged.

**Figure 11 jcm-13-01516-f011:**
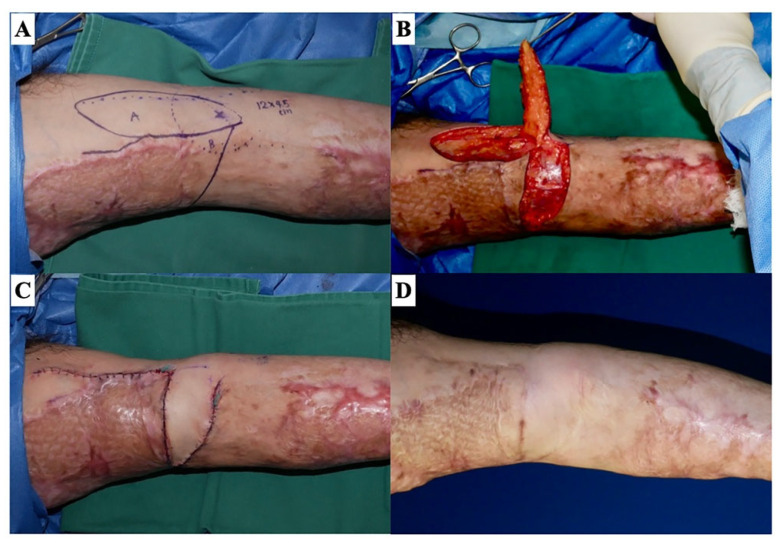
Type 2-s: planar scar contracture confined to the superficial layer. A 47-year-old man with a history of flame burn has a right upper arm scar contracture. (**A**) The design of a transposition flap. (**B**) Intraoperative photographs show that the perforator is included in the flap pedicle. (**C**) Immediately after suturing. (**D**) The normal skin sandwiched between the scar contracture lines becomes wider over time.

**Figure 12 jcm-13-01516-f012:**
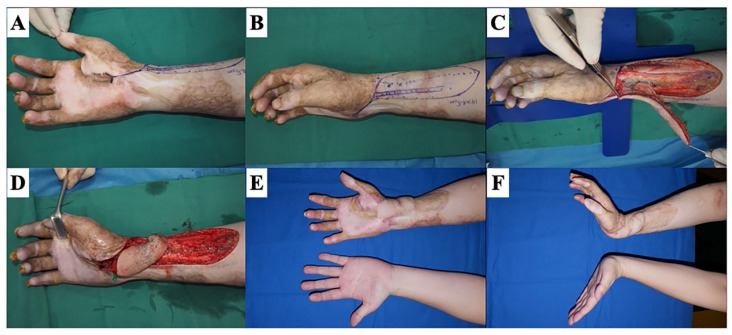
Type 3: planar scar contractures reach the deep layer (3D scar contractures). A 29-year-old male with a history of flame burn has a wrist contracture. (**A**) The scar contracture ranges from the wrist to the thumb. (**B**) The design of a radial artery perforator flap. (**C**) During flap elevation. (**D**) The flap was transposed to cover the defect after releasing the scar contracture and underlying transverse carpal ligament. (**E**,**F**) Twelve months after the surgery: no limitation of the wrist extension.

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
