# Peer review of "Management of Scar Contractures of the Hand—Our Therapeutic Strategy and Challenges"

_jcm, 2024, doi:10.3390/jcm13051516_

Round 1

Reviewer 1 Report

Comments and Suggestions for Authors

Review Report

Journal: JCM (ISSN 2077-0383)

Manuscript ID: jcm-2833836

Type: Article

Title

Management of Scar Contracture of the Hand – Our Therapeutic Strategy and Challenges –

Authors

Hoyu Cho , Shimpei Ono * , Kevin C Chung

Disclosure Statement: I have no potential conflicts of interests.

Dear Authors,

Thank you very much for the submission of your article about “Management of Scar Contracture of the Hand – Our Therapeutic Strategy and Challenges”. In your article, you present the challenges of scar management especially in the case of scar contractures of the hand. You provide a method of clinical assessment and use this to guide possible procedures for covering the defect. This new classification is intended to make decision-making easier.

Some revisions must be carried out:

1.)    You should integrate the possibility of dermal substitutes such as Integra, Novosorb®BTM, MatriDerm especially in complicated wounds or after various attempts of defect coverage. It is very promising and would add you article more depth.

Examples:

Frame JD, Still J, Lakhel-LeCoadou A, Carstens MH, Lorenz C, Orlet H, Spence R, Berger AC, Dantzer E, Burd A. Use of dermal regeneration template in contracture release procedures: a multicenter evaluation. Plast Reconstr Surg. 2004 Apr 15;113(5):1330-8. doi: 10.1097/01.

Ellis CV, Kulber DA. Acellular dermal matrices in hand reconstruction. Plast Reconstr Surg. 2012 Nov;130(5 Suppl 2):256S-269S. doi: 10.1097

2.)    As interesting as your article is, I don't understand the usefulness of your classification. What is the added value for the surgeon? The classification does not provide any new information about defect coverage. Your statements about the various flaps or defect analysis have been known for a long time and have been described several times. There is no novelty other than a further classification. You should write the article as a more thorough review about scar contractures and its surgical management with clinical tips and tricks and possible pitfalls without giving an emphasis on a new classification.

Author Response

Response to comment 1):

Thank you for your constructive comment and for showing the example papers. As you mentioned, the acellular dermal matrix is useful for severe scar contracture cases because they rarely have a healthy donor surrounding the defect after releasing the scar contracture. We changed and added sentences as below in lines 260-267 of the new manuscript. Please see the attachment.

In cases of circumferential or multiple and extensive burns of the extremities and joints, the area of scar contracture is large and there are few surrounding healthy donor-site to cover the defect after releasing the scar. The paucity of donors is particularly problematic in pediatric cases. The risk of recurrence of scar contracture is inversely proportional to the dermis amount and wound bed condition is closely related to the quality of treatment of scar contracture.15,16 The use of artificial dermis for wound bed preparation before skin graft provides good outcomes in terms of reducing the recurrence rate of scar contractures and increasing the range of motion of the joints.16,17,18

Response to comment 2):

Thank you very much for taking the time to review this article and for your comment. As you noted, this may seem obvious to well-experienced physicians. However, for physicians with less experience, this classification would be beneficial in determining the treatment strategy. As far as we could find, there are no charts that allow for morphological classification of scar contractures and treatment options.

Specifically, identifying the surgical options with this simple chart would allow physicians with less experience to be more proactive in instituting scar contracture optimization strategies, while also being more aggressive in the treatment of scar contracture. Therefore, this paper would be of benefit to the reader.

Reviewer 2 Report

Comments and Suggestions for Authors

The author reported the strategies of hand contracture treatment.

The manuscript is well written and easy to understand.

I have a question.

The author mentioned the classification for the surgical treatment (type1, 2-d, 2-s and 3).

Can we categorize to this classification preoperatively?

If this classification is based on the surgical findings, it should be written in limitation.

And I guess the distinguish between type1, 2-s and 2-d, 3 might be obscure.

Author Response

Thank you for your comment and we appreciate you taking the time to review this manuscript. To distinguish the dimensional classification, evaluation by actual palpation is important and we consider it possible to evaluate preoperatively. Scar tissue does not always reach flat, and there is a mixture of areas that reach a superficial layer and areas that extend deeper. Therefore, as you mentioned, there are some scars whose classification is ambiguous. However, the purpose of this manuscript is to present a systematic treatment strategy in a simple procedure. We changed and added sentences as below in lines 177-183 and 268-270 of the new manuscript. Please see the attachment.

In lines 177-183

To distinguish the dimensional classification, preoperative evaluation by actual palpation is important. When the scar reaches deep enough to cause adhesions to the underlayers, the range of motion will be limited. Therefore, it could be possible to estimate the range of the scar by using passive extension, flexion, and pinching as shown in Figure 3. Scar tissue does not reach flat, and there is a mixture of areas that reach a superficial layer and areas that extend deeper. However, this is considered to be a simple procedure for determining the surgical strategy.

In lines 268-270

In any case, it is preferred to approach the surgery with the assumption that the situation is one rank more severe as if the preoperative evaluation led the surgeon to believe it was type 1 and releasing the scar, which turned out to be type 2 with a wider defect.

Reviewer 3 Report

Comments and Suggestions for Authors

Thank you for letting me review this article.

The topic is very interesting, and the information given can be beneficial for all hand surgeons.

My concern about this manuscript is that it looks more like a book chapter than a scientific article.

The bibliography should be implemented; this article is full of affirmations without any citations. This article has no scientific soundness, although the content is interesting. I suggest modifying the design of the study, presenting some results of your works, and comparing those results to other works.

Author Response

Thank you for your comment and we appreciate you taking the time to review this manuscript. This is a review paper, not an original article, and we have applied for and received a change of registration regarding the change of article type. We apologize for any inconvenience this may cause and hope you will understand. We have attached the new manuscript reflecting the opinions of the other reviewers. Please see the attachment.

Round 2

Reviewer 1 Report

Comments and Suggestions for Authors

Dear Authors,

thank you very much for your point-to-point answers and additions. The manuscript reads now more rounded and complete. Although I have a different opinion about your classification, I can see your point.

I no longer have any concerns about publishing your manuscript.

Best of luck and continued success.

Reviewer 3 Report

Comments and Suggestions for Authors

Thank you for the revisions and for having clarified the type of paper this is.